# Informing Patients about Biosimilar Medicines: The Role of European Patient Associations

**DOI:** 10.3390/ph14020117

**Published:** 2021-02-04

**Authors:** Yannick Vandenplas, Steven Simoens, Philippe Van Wilder, Arnold G. Vulto, Isabelle Huys

**Affiliations:** 1Department of Pharmaceutical and Pharmacological Sciences, KU Leuven, 3000 Leuven, Belgium; steven.simoens@kuleuven.be (S.S.); a.vulto@gmail.com (A.G.V.); Isabelle.huys@kuleuven.be (I.H.); 2Ecole de Santé Publique, Université Libre de Bruxelles (ULB), 1050 Brussels, Belgium; Philippe.Van.Wilder@ulb.be; 3Hospital Pharmacy, Erasmus University Medical Center, 3015 GD Rotterdam, The Netherlands

**Keywords:** biosimilar, biological, information, education, communication, patient, Europe

## Abstract

Biosimilar medicines support the sustainability of national healthcare systems, by reducing costs of biological therapies through increased competition. However, their adoption into clinical practice largely depends on the acceptance of healthcare providers and patients. Patients are different from health care professionals (HCPs), who are informing themselves professionally. For patients, the biosimilar debate only becomes actual when they are confronted with disease and drug choices. This paper provides a literature review on how patients are and should be informed about biosimilars, searching in scientific databases (i.e., Medline, Embase). Several large surveys have shown a lack of knowledge and trust in biosimilars among European patients in recent years. This review identified five main strategies to inform patients about biosimilars: (1) provide understandable information, (2) in a positive and transparent way, (3) tailored to the individual’s needs, (4) with one voice, and (5) supported by audiovisual material. Moreover, the importance of a multistakeholder approach was underlined by describing the role of each stakeholder. Patients are a large and diffuse target group to be reached by educational programs. Therefore, patient associations have become increasingly important in correctly informing patients about biosimilar medicines. This has led to widespread biosimilar information for patients among European patient associations. Therefore, a web-based screening of European Patients’ Forum (EPF) and International Alliance of Patients’ Organizations (IAPO) member organizations on publicly available information about biosimilars was performed. We found that the level of detail, correctness, and the tone of the provided information varied. In conclusion, it is paramount to set up a close collaboration between all stakeholders to communicate, develop, and disseminate factual information about biosimilars for patients.

## 1. Introduction

Since their introduction to the European market in 2006, biosimilar medicines have contributed to a more sustainable healthcare system in several European markets [1]. Biosimilars are biological medicines that contain a version of the active substance of an already authorized biological medicine in the European Economic Area (EEA) [2]. They are allowed to enter the market when market exclusivities of the original biological product have expired, and market authorization has been granted by the European Commission (EC). Market authorization is achieved after a rigorous regulatory evaluation process by the European Medicines Agency (EMA) and subsequent approval of the EC. This guarantees that biosimilars are as effective and safe as their reference product, making them equal treatment options for patients [2,3]. Several benefits have been identified following the increased competition induced by biosimilar market entry [4]. Due to the decreased costs of biological medicines, generated savings could be allocated to providing patients with more access to biological therapies. In addition, these savings can be utilized to finance high-cost innovative treatments [1,5,6].

However, the extent to which these benefits are being captured in Europe largely depends on the adoption of biosimilars by European Union (EU) member states. Adoption into clinical practice might be hampered by limited healthcare provider (HCP) and patient acceptance of biosimilars. Often, besides other factors such as the absence of tangible incentives, a lack of acceptance among HCPs and patients comes down to shortcomings in knowledge and understanding about biosimilars [7,8,9]. Patients’ access to information and education about biosimilar medicines is therefore considered as one of the key elements for a sustainable market [10]. Hence, policy initiatives aiming to increase understanding among clinicians and patients have been implemented in most European countries in past years [11,12].

Several studies have brought an inadequate understanding and acceptance among European patients about biosimilars to light, underlining the need for information and education of patients [13,14,15,16,17,18,19]. Especially when transitioning or *switching* current original (or innovator) biological therapy to its biosimilar, the value of adequate patients’ understanding about biosimilars cannot be underestimated [20]. Clinical studies have proven the positive effect on patient outcomes when patients with rheumatological disorders were properly informed before transitioning to a biosimilar [21,22]. The authors attributed the improvement in patient outcomes after a structured communication strategy to a reduction in the risk of *nocebo effects.* The nocebo effect is described as the worsening of symptoms associated or an increase in side effects with a negative attitude towards a given therapy, in this case the biosimilar medicine. A lack of patient knowledge is the main underlying reason for negative attitudes towards biosimilars, contributing to nocebo effects and possible treatment failure [20,23].

Educating patients about biosimilars is crucial to provide clarity and prevent misinformation [9,20,24]. Patients need access to understandable and evidence-based information that allows them to make informed decisions about their treatment. Regulatory authorities, medical scientific associations, and patient organizations have therefore been active in developing and disseminating educational material on biosimilars for European patients during past years. However, information and educational material are widespread, requiring a mapping of the available material [8,9]. Mapping the available information or material for patients makes it possible to have an overview of what material exists, and to verify the information found for its scientific correctness. In addition, a proper inventory will facilitate the dissemination of information through collaboration between stakeholders.

This review aimed to provide an overview of existing scientific literature on how to inform patients about biosimilars and compile available information about biosimilars for patients, developed or disseminated by European patient associations. Based on this review, an overview of the important aspects when talking to patients about biosimilars is provided for policymakers, healthcare providers, patient organizations, and other relevant stakeholders, in support of a sustainable market for off-patent biological and biosimilar medicines in Europe.

## 2. Methods

### 2.1. Literature Review

This comprehensive structured literature review identified articles on what information patients need about biosimilars and how this information can be communicated, by looking into scientific databases (Embase, Medline) using a structured search strategy (Cfr. Appendix A). Relevant English-language scientific publications published between 2006 and 2020 were included. This period was chosen since biosimilars have been introduced in Europe in 2006, thereby encompassing the whole period of time when biosimilars were available on the European market. Search terms were related to patient communication about biosimilars and included the following terms: ‘biosimilar’, ‘information’, ‘education’, ‘communication’, ‘knowledge’, and ‘patient’. All terms were modified according to the respective scientific database. Both abstracts and full texts were included in the analysis. Only articles relevant to the European landscape were within the scope of this analysis. Articles were searched up to the 21st of October 2020.

All identified records were imported from Embase or Pubmed into Mendeley software to remove duplicates. Next, all articles were screened on title and abstract for relevance in the Rayyan (Qatar Computing Research Institute, Doha, Qatar) software. In a third step, articles were carefully reviewed based on their full text. Lastly, reference lists of included articles were searched for additional relevant articles. The articles included in the final analysis were analyzed qualitatively according to the thematic framework method [25]. A combination of inductive and deductive coding was used, since some aspects were already identified as relevant for this research question. During the initial coding step, general themes were identified prior to the literature review. Similar codes were grouped together to form the coding tree. Second, the identified literature was coded deductively. Meanwhile, additional codes were created inductively and added to the coding tree.

### 2.2. Mapping of Patient Information

A web-based screening on relevant patient information (i.e., general information not intended for educational purposes) or educational material (i.e., brochures, toolboxes, position papers, audiovisual material, etc.) was performed to provide an overview of the educational material disseminated by European patient organizations. This screening included all public websites of European Patients’ Forum (EPF) and European International Alliance of Patients’ Organizations (IAPO) members. EPF and IAPO are two major umbrella associations, uniting a large number of European patient organizations in a variety of disease areas. Websites were screened on available information about biosimilars by searching for ‘biosimilar’ or related terms in the search bar. In addition, the name of the respective patient association was combined with the term ‘biosimilar’ via Google to make sure no information was missed.

After all identified information was analyzed and mapped together, the tone in which each association reports about biosimilars was evaluated on a five-point Likert scale. This was done by scoring the overall attitude towards biosimilar medicines on the following scale: “− −” (negative), “−” (somewhat negative), “0” (neutral), “+” (somewhat positive), “+ +” (positive). Neutral information was taken as a starting point. Neutral information refers to factually correct information about biosimilars, without any additional positive or negative undertone. The initial scoring was done by one researcher (Y.V.), and afterwards reviewed by four other researchers (S.S., A.G.V., P.V.W., I.H.).

The purpose of the web-based screening was (1) to examine to what extent patient information about biosimilars is provided on their public websites, (2) to have a closer look at the actual content of these materials, and (3) to evaluate the tone in which they report about biosimilars. All different types of information found was schematically listed per patient association (Table 1).

## 3. Results

### 3.1. Literature Review

After a screening of 1319 records, a total of 51 articles were included in this literature review. Although conference abstracts (*n* = 6) were also eligible for inclusion, most identified records were full-text articles (*n* = 45). Most articles were identified through the structured literature search after title and abstract screening (*n* = 38). Nonetheless, the screening of reference lists resulted in 13 additional records. A complete overview of the literature search process is included in Appendix A. 

#### 3.1.1. Points to Consider When Talking to Patients about Biosimilars

In the vast body of literature, we can conclude that several specific aspects are essential when informing patients about biosimilars. An overview of these aspects is provided below.

##### Provide Understandable and Up-to-Date Information

Biosimilars are a relatively new and difficult concept for patients. It is therefore important that the given information to patients is easy to understand and not overly complicated. The message must be concise, using simple language, avoiding redundant medical jargon [50,51,52]. When informing the patient face-to-face, make sure they understand all information by asking questions and involving them in the discussion [52]. In this way, the patient will feel more involved and can participate in the discussion as well. In addition, the information must be up-to-date and adapted to the most recent insights [53]. It should not contain outdated concepts or outdated data.

##### Communicate Positively

Several studies have already shown that it is crucial to positively formulate the message about biosimilars towards patients. An empathic and positive communication (including positive framing) or attitude increase the acceptance to switch and reduce the development of nocebo effects after transitioning to a biosimilar [54,55,56,57,58]. An open and positive communication, emphasizing the equalities and not the differences between the reference product and its biosimilar, should be the norm when talking to patients. Information or communication should avoid messages such as: “biosimilars have no meaningful differences with their reference product”. Instead, the similarities must be underlined in any communication to reassure patients that biosimilars are equal treatment alternatives [20,58,59]. When transitioning to a biosimilar, it is unnecessary to mention all possible side effects. It is rather recommended to provide patients with the opportunity to contact their physicians or nurse when any unexpected side effect would occur [60]. Moreover, a positive communication about biosimilars should be adopted for information towards HCPs as well, thereby supporting overall acceptance of biosimilars in clinical practice [9,50].

Patients generally feel that their physician’s opinion and attitude on biosimilars strongly influences their decision to use a biosimilar [61]. Yet, an open and positive attitude should be adopted by all healthcare providers (i.e., physicians, nurses, and pharmacists) who communicate with patients. This involves empathy, reassurance, and nonverbal elements in their communication towards patients when discussing medicines in general [51,62]. It will be essential to educate HCPs using these communication techniques or ‘soft skills’ in the future.

##### Provide Information Tailored to the Individual Patients’ Needs

A one-size-fits-all approach to communicate or inform patients about biosimilars does not exist, nor would it be appropriate [8,63]. Some patients will naturally be more concerned about their treatment and ask for more information. While other patients trust their physician completely and will express no further concerns about biosimilars [55,60,64]. However, many patients will be somewhere between these two extremes of the spectrum, highlighting the importance of tailored communication. Providing too much information could lead to unnecessary concerns of patients, while giving too little information could leave patients with remaining concerns [65]. It is the task of all HCPs to assess the individual patients’ needs and find the right balance. Specific tools or questionnaires exist to assess prior beliefs or concerns of patients about their medicine, such as the Beliefs about Medicine Questionnaire (BMQ) [63]. The BMQ might help HCPs stratify patients based on their prior thoughts about biosimilars before transitioning.

In addition, information should be tailored to the individual patient’s demographics and health literacy as well [66]. For example, patients affiliated to a patient association or previously treated with a biological medicine generally have a better knowledge about biosimilars [16]. Some patients might have already looked for information about biosimilars elsewhere, given the broad access to information on the internet [64,67]. It is therefore advised to account for this and assess whether their prior knowledge is factual. Furthermore, in order to make sure that the information is accessible for all patients, educational material should be translated into local languages.

##### Communicate with One Voice

As already touched upon in the above, communication towards patients must be consistent across resources, so confusion among patients is avoided. Homogenous information leads to higher acceptance and better treatment outcomes after transitioning to a biosimilar [54,57]. Stakeholders should therefore deliver the same message or speak with *one voice* to patients about biosimilars [7,20]. Such an approach means that all healthcare providers are involved and educated about biosimilars, ensuring a coherent and unified message to patients. Not only the information itself, but also the way it is explained to patients should be coherent (i.e., positive and open communication, tailored information) [68].

##### Make Use of Supportive Material

Several ways exist to inform patients in addition to oral communication of the HCP with the patient. In the context of transitioning or switching to biosimilars, written informed consent before transitioning could be considered. Such information must be in the patient’s native language, include only key information on biosimilars, the reasons why transitioning is considered, and who to contact if they have any issues or concerns [50,54,57,69].

For general information accessible to patients, a variety of audiovisual aids can be used, such as videos, infographics, podcasts, and pictures [50,52]. All these ways may contribute to the understandability and confidence in the key biosimilar concepts. Moreover, for subcutaneous biosimilars, instructional leaflets or videos about the injection device might be useful as well. Since patients are increasingly seeking health-related information on the internet, such audiovisual material can be made broadly accessible online [67]. For example, the European Medicines Agency (EMA) and European Commission (EC) developed an animated video explaining the general concepts of biosimilar medicines [70].

#### 3.1.2. Information Needs of Patients about Biosimilar Medicines

A multitude of studies has been performed in past years assessing the level of knowledge about or attitudes towards biosimilar medicines among European patients. In general, most of these studies concluded that the level of knowledge of patients is limited, as well as that confidence in biosimilars is rather low. In particular, limited knowledge about the general concepts of biological and biosimilar medicines is reported [13,14,15,16,17,19,54,56,71,72,73]. Doubts around efficacy, safety, and extrapolation of indications were revealed among most patient populations (i.e., oncology, psoriasis, rheumatology, IBD). It goes without saying that correct information and education can resolve these concerns and lack of knowledge.

A tailored approach was already pointed out earlier in this review in the context of direct communication of HCPs towards patients. The specific biosimilar concepts that should be explained by HCPs will therefore vary from patient to patient, depending on the individual needs and level of understanding. It used to be common practice that the basic concepts about biological medicines, and biosimilars in particular (e.g., definitions, safety, efficacy, regulatory approval, etc.), have to be clearly explained to patients when transitioning to a biosimilar [20,60,74]. However, nowadays current practice has evolved towards providing the message that another brand of the same medicine will be used, with the same efficacy and safety outcomes at a lower cost.

There is still a lack of clarity about which aspects of biosimilars should be included when developing educational material for patients [8]. It should be borne in mind that patients themselves look for information about biosimilars on the internet, potentially finding incorrect information. The purpose of providing information is to counter such negative reports as well [9,74,75]. Therefore, publicly available information or educational material about biosimilars for patients should address the general definitions of biological and biosimilar medicines in an understandable way. This should include the thorough regulatory evaluation process of EMA that assures the same clinical efficacy and safety between the original and biosimilar product. The potential benefits of biosimilars can also be considered, albeit in understandable language and as direct benefits (i.e., increase in access to necessary medicines or access to treatments at an earlier disease stage) [8,54,76]. However, it should be avoided that the impression is created that the patient is treated with biosimilars only for the sake of cost savings. Other essential concepts such as extrapolation of indication may be explained as well, although overly detailed information should always be avoided [8,20].

#### 3.1.3. Reaching the Patient

All stakeholders, particularly healthcare providers, play a role in informing patients about biosimilars. It must be stressed that communicating with patients should be a multistakeholder effort [8,20,77]. This includes physicians, nurses, pharmacists, scientific associations, regulatory bodies, and patient associations. In the following, we summarize the role of each stakeholder in informing patients about biosimilars (Figure 1).

##### Role of Physicians

Treatment decisions must be based on shared decision-making between patients and their physician. In most European countries, physicians have the ultimate responsibility in making treatment choices. Physicians will often be the first point of contact for patients when treatment decisions are being made, and they should therefore ensure a trusted relationship with the patient. Good communication, based on informed discussions and shared decision-making with the physician, is known to benefit adherence to a prescribed medicine, and thus the adoption of biosimilars [5,20,21,74]. However, shared-decision making about medical therapy in general is not yet established to the same extent in every European country [78,79].

Previous research involving patient surveys has shown that physicians are the most trusted source of information about biosimilars [17,19,80]. However, several surveys among European physicians have concluded that physicians’ knowledge on biosimilar medicines could be improved [50]. As a result, it is clear that physicians should be properly trained about biosimilars and be able to communicate adequately about them to the patient. As mentioned earlier, physicians must therefore be trained in communication techniques as well [65].

##### Role of Nurses

Nurses play a key role in the daily care for patients and are ideally placed to inform patients by addressing questions or concerns about their medicine. Usually, nurses administer the medication and spend the most time with patients, which allows them to have a closer relationship with the patient [81]. When transitioning from a reference product to its biosimilar, the important role of nurses has been pointed out in several publications during past years [7,8,52,81,82]. Building further on their profound experience with educating patients, nurses can guide patients in the process when transitioning to a biosimilar and manage nocebo effects. Additionally, following the transition or initiation with a biosimilar, patients may have further questions or concerns at home. To prevent any additional concerns or even discontinuation of their treatment, nurses should serve as a contact point to patients [58,83]. For subcutaneously administered biologicals, where injection devices may differ, nurses provide the necessary explanation and guidance to use the new injection device [52].

The above reasons make it clear that nurses are a critical link in the multidisciplinary team, particularly when making the transition to a biosimilar. This has been recognized by the European Specialist Nurses Organization (ESNO), by developing an elaborate communication guide for nurses when transitioning to a biosimilar in 2017 [84]. This document has been translated into eight languages and can serve as a reference document for nurses.

##### Role of Pharmacists

The main task of a pharmacist is often simplified to the delivery of medicines. However, pharmacists also have an important task of providing information to patients, although regional differences exist among European countries in their role in direct patient counseling. Especially community pharmacists serve as a first-line contact for patients for any questions about their medicine, including biosimilar medicines [58,66]. Pharmacists thereby contribute to medication adherence by increasing confidence in biosimilar medicines among patients. They may also have to explain differences in injection devices, since subcutaneously administered biosimilars are often dispensed in community or outpatient pharmacies.

For biosimilars delivered in the hospital setting, pharmacists have an increasing role in educating the medical staff about biosimilars [85,86]. The Dutch association of hospital pharmacists (NVZA) has developed a practical guidance document (i.e., toolbox) on how to implement biosimilars in the hospital setting, thereby emphasizing the role of hospital pharmacists in this process [87]. As medicine experts, clinical pharmacists can serve as a coordinator of the medical team to address patients’ concerns about biosimilars when preparing the switch to a biosimilar. Their role should be further explored in the future, particularly in the context of transitioning to biosimilars in the hospital setting.

##### Role of Scientific or Medical Associations

Several European scientific associations have developed educational material for patients in past years about biosimilars. Due to their extensive scientific expertise and background, they are an important source of unbiased information about the use of biosimilars [62,66,71]. The European Society for Medical Oncology (ESMO) developed educational leaflets about biosimilars for patients [88]. ESMO uses infographics to explain the key concepts and potential advantages of biosimilars in understandable language. The European League Against Rheumatism (EULAR) has developed a document with general information about biosimilars as well. The main questions or concerns patients may have are addressed in this question and answer brochure [89,90]. Additionally, the need for more patient educational material is highlighted in this document.

##### Role of Regulatory Authorities

European regulatory agencies and national competent authorities have a supporting role in disseminating unbiased information about biosimilars in general [55,66,91]. However, room for improvement was recently pointed out for European national competent authorities to disseminate biosimilar information to the public [92]. The widespread patient brochure developed by the European Commission (EC) and European Medicines Agency (EMA) has become a reference document for patients, and is being referred to by many national authorities [93]. This brochure was developed in cooperation with the European Patients’ Forum (EPF) in 2016, explaining the key concepts about biological and biosimilar medicines in lay language. It is also publicly available in a more concise video format [70]. In recent years, this material has been translated into all European languages [93]. National authorities should continue facilitating the dissemination of this document, as it provides coherent and factual information about biosimilars in understandable language and graphical format [92].

##### Role of Patient Associations

Patient organizations are a trusted source of information for patients about biosimilars. Patients rely on their respective associations or advocacy groups to clarify complex concepts such as biosimilars [16,19]. Patient associations can also serve as a discussion board to discuss complex matters such as biosimilars and share experiences among patients [17]. If patient associations are committed to developing educational material themselves, they should join forces with medical and scientific associations. In this way, it can be ensured that the information is evidence-based and up-to-date [13,71]. 

A schematic overview of the multistakeholder approach, using the five identified strategies, is provided in Figure 1. The section below takes a closer look at the role of patient associations in developing and disseminating information about biosimilars to patients.

### 3.2. Information Provided by European Patient Organizations

In total, public websites of 75 European Patients’ Forum (EPF) members and 95 members of the International Alliance of Patients’ Organizations (IAPO) were consulted. As some organizations were part of both EPF and IAPO, 159 unique members were screened. Of these 159 patient organizations, 16 were actively disseminating information on biosimilars via their website. An overview summarizing all patient organizations, along with the type of information, is provided in Table 1.

Patient associations active in providing information about biosimilars are representing patients with a variety of diseases or regions. The main disease areas are those where biosimilars are marketed today, such as rheumatology, diabetes, oncology, inflammatory bowel diseases, and psoriasis. The majority of these associations only provide brief information on biosimilars, by explaining key concepts or merely providing a link to the patient brochure developed by the European Commission (EC) [93]. Nonetheless, some patient organizations have developed their own educational material or even produced position statements on the use of biosimilar medicines within their specific disease area. All identified information or educational material on biosimilars intended for patients is summarized below (Table 1).

The Slovakian Association for the Protection of Patients’ Rights (AOPP) provides a short article briefly explaining the main characteristics of originator biological and biosimilar medicines. For more information, they refer patients to the EC brochure [26].

One of the larger patient associations discussed in this review is Digestive Cancers Europe (DiCE). It is the umbrella organization of a larger group or national associations representing patients with colorectal, gastric, and pancreatic cancers. DiCE has committed itself in recent years to several educational initiatives. In 2019, they developed a position paper on biosimilars for the treatment of colorectal cancer [27]. In this well-structured paper, they touch on the definitions of biologicals, with specific information about biosimilar medicines. They also draw attention to the benefits of biosimilar usage, in particular the increase in access to biological medicines. Problems regarding unequal access to biologicals among European countries are mentioned, including the possible role of biosimilars to overcome these to a certain extent. More recently, in the context of the licensing of bevacizumab biosimilars in Europe, DiCE started a larger project to provide educational material about biosimilars for patients and HCPs [28].

Similar to DiCE, the European Federation of Crohn’s and Ulcerative Colitis Associations (EFCCA) is the umbrella organization representing national Crohn’s and ulcerative colitis patient associations. Like most patient associations discussed in this review, they mention the EC brochure about biosimilars for patients. EFCCA also wrote an article about biosimilars, mentioning specific information on biosimilars and their benefits for healthcare systems in their monthly magazine. In this article, EFCCA emphasizes the importance of generics and biosimilars for a competitive market and a more sustainable healthcare system. In addition, they state that physicians should not be obliged to prescribe a biosimilar purely on the grounds of cost, but should be allowed to exercise appropriate clinical judgment and always involve patients in the decision making process [31].

Even though no biosimilars have been marketed yet for the treatment of Parkinson’s Disease, the European Parkinson’s Disease Association (EPDA) provides a brief explanation of biologicals in general, as well as of biosimilar medicines. They emphasize that all biological medicines are prone to structural variability and the possible consequences on clinical outcomes. Due to the varying composition of biologicals, patient safety may be a concern. The only specific information given on biosimilars is that they aim for the same mechanism of action as the original, even though different cells are used during the production process [33].

The International Diabetes Federation Europe or IDF Europe is the umbrella organization of European national associations for patients with diabetes. IDF Europe has developed a position paper on the use of biosimilars among patients with diabetes in 2017 [34]. In this extensive position document, several topics are highlighted such as the difference between biosimilars and generics, the European legal framework, and the potential impact of biosimilars on healthcare systems. The paper ends with a set of recommendations for the use of biosimilars in clinical practice. Under the list of recommendations, IDF Europe states that stable patients on insulin treatment should not be switched to a biosimilar without good clinical reasons and evidence of interchangeability. Furthermore, patients should always be informed and involved in the decision-making process, based on an informed discussion with their physician. They demand more information for patients from national regulatory authorities, specifically about biosimilar medicines. Routine education for patients with diabetes, facilitated by national authorities, should include a section on biosimilars. However, the position paper emphasizes possible clinical differences between insulin biosimilars and reference products. According to the authors, not enough clinical evidence exists to ensure biosimilars are equally safe and effective as their reference product. Possible immunogenicity risks are pointed out, especially when switching the reference biological with its biosimilar. To support this statement, they refer to an epoetin biosimilar (HX-575) that showed an increased occurrence of adverse events linked to a higher immunogenicity of the biosimilar. However, the article they refer to does not mention a possible difference between the biosimilar and originator of epoetin due to increased immunogenicity [94]. Instead, the article describes several cases of pure red cell aplasia (PRCA) with epoetin treatment, among which a trial with a biosimilar of epoetin. The particular clinical study being referred to reported two cases of neutralizing antibodies with the epoetin biosimilar [95]. An extensive analysis revealed that contamination during primary packaging of the prefilled syringes explained the increase in neutralizing antibodies [96]. The manufacturing process was therefore improved, followed by the completion of new open-label study without any patients developing neutralizing antibodies. Subsequently, the respective biosimilar HX-575 was authorized on the European market in 2016. This type of information is an example of a false narrative, by supporting incorrect conclusions with references from published scientific articles. Such kind of incorrect and negatively framed information should be avoided, since it may harm the trust in biosimilar medicines among patients with diabetes and potentially lead to a slower adoption of biosimilars [9].

Malta Health Network (MHN), the national association for Maltese patients, provides a link to the EUPATI toolbox on biosimilar medicines [35]. Although this information is rather difficult to find on the MHN website, the EUPATI toolbox provides understandable information on biological medicines, including biosimilars, for patients [97].

The Spanish Platform for Patient Organizations published an article for patients about biological and biosimilar medicines in 2017 [36]. Definitions about biologicals, biosimilars, and the difference with generic medicines are highlighted. They underline the importance of therapeutic freedom of physicians when prescribing biosimilars, and switching must always be in close dialogue with the patient. While they are not opposed to switching the original product with the biosimilar, they state that there is insufficient evidence to support switching.

Another national patient association is the National Coalition of Dutch Patients, which provides simple and understandable information about biosimilars on its website [39]. Like many other patient associations discussed in this review, the importance of involving the patient in the decision to prescribe a biosimilar is mentioned. For further information, they refer patients to a brochure developed by the Dutch competent authority [40]. This is a structured document providing information about biological and biosimilar medicines in understandable language for patients. Moreover, the question and answer structure of the document might increase the understandability of the brochure. In contrast to other informational material discussed in this article, the same effect of the biosimilar and the reference product is emphasized instead of no expected differences. In general, a more favorable position towards switching to a biosimilar is noted. Yet, switching must remain the physician’s responsibility, and the necessary consultation with the patient is required.

The Flemish Patient Platform (FPP) unites Dutch-speaking Belgian patient associations. FPP mentions very limited information on biosimilars, merely explaining that they are biological medicines. FPP refers to biosimilars as copies, similar to generic medicines, which is too simplistic and incorrect. Furthermore, they advise patients to look at the Belgian national competent authority’s information on biosimilars [42]. Here, general information about biosimilars is provided, including definitions, approved biosimilars, and guidance when switching to biosimilars. However, this web page is not up to date and only contains information of approved biosimilars until 2016.

The European Multiple Sclerosis Platform (EMSP), European Federation of Neurological Associations (EFNA), European Institute of Women’s Health (EIWH), and European Patients’ Forum (EPF) only posted the link to the EC brochure for patients on their website [32,37,38,43].

Three additional patient organization members of IAPO were identified that provide educational material for patients on their website. The International Federation of Psoriasis Associations (IFPA) is the overarching organization of national patient associations representing patients with psoriasis. IFPA recently developed a position paper about the use of biosimilars for the treatment of psoriasis [45]. They acknowledge that biosimilars do not lead to different clinical outcomes compared with their reference product. Again, the patient–physician dialogue is underlined when making treatment decisions in general, which includes decisions to switch to a biosimilar. However, they mention transitioning to a biosimilar should not be done for patients with stable disease control. This shows some hesitance to use biosimilars among patients with psoriasis already treated by biological medicines.

Psoriasis Action, the Spanish patient association for patients with psoriasis, provides several sources of biosimilar information. They share a short video in which a professor explains what biosimilars are to Spanish patients [47]. A video with a more extensive explanation is also provided, intended for patients who prefer more detailed information on biosimilars [48]. Psoriasis Action also published a more general piece of information for patients, where generalities of biological and biosimilar medicines are explained in a specific article [46].

Last but not least, IAPO also developed educational material for patients about biosimilars in collaboration with the International Federation of Pharmaceutical Manufacturers and Associations (IFPMA). On their website, extensive documentation on biological and biosimilar medicines can be found in their toolkit for patients, from which they developed a second version in 2017 [49]. The toolkit includes fact sheets, infographics, frequently asked questions, and a decision guide for patients when choosing between an original biological or biosimilar product. Their educational material includes general information on biological and biosimilar medicines, regulatory requirements, pharmacovigilance, and a communication guide for HCPs. The toolkit is intended for patient organizations worldwide for distribution to their patients. In contrast to the initial version of this toolkit of 2013, which was made available in English, Spanish, and Portuguese, the second version is only available in English [98].

## 4. Discussion

This article looked at the relevant elements to consider when informing patients about biosimilars. In addition, an overview of the information and educational material by the major European patient associations was provided. Based on this overview, all available material was evaluated on its tone and correctness.

### 4.1. Communication Strategies to Inform Patients about Biosimilars

Five main points of attention were identified when informing patients about biosimilars. **First** of all, information has to be provided in an understandable way. Patients generally have no scientific background, so one must make sure not to overly complicate the given message [50,51,52,53]. **Second**, a positive attitude when talking to patients about medicines in general is paramount [20,51,58,62]. Emphasis must be put on the similarities between biosimilars and their reference product, rather than the possible differences. This can be done by conveying the message that the biosimilar has similar clinical outcomes, instead of no expected differences [59]. An open and positive way of communicating has shown to generate trust, and subsequently improve treatment outcomes and adherence [56,57]. HCPs should therefore be trained on the proper use of such communication strategies with patients. **Third**, a one-size-fits-all approach is not desirable when communicating directly to patients since each patient’s individual needs and level of understanding might differ [8,60,63,64,66]. A tailored approach is therefore preferred. It is the task of each member of the multidisciplinary team to assess these needs and to adapt their communication strategy accordingly. This brings us to the **fourth** point of attention, the *one voice* principle. In essence, this means that everyone informing patients about biosimilars has to provide a coherent message. Communication towards patients must be consistent across channels, thereby avoiding suspicion by generating trust between healthcare providers and patients [7,20,54,58,68]. **Fifth**, the use of supportive audiovisual material (i.e., videos, infographics, brochures) may help bringing the information across in a clear and understandable way [9,50,52,67]. Such supportive material closes the gap between the complexity of the biosimilar concepts and the need for understandable information.

A series of studies pointed to a lack of knowledge and trust in biosimilars in various relevant patient populations, making clear the necessity of education [13,14,15,16,17,19,54,56,71,72,73]. However, the purpose of informing patients should not be to create a high level of knowledge among the whole patient population. This would not be feasible, nor desirable. After all, it is not intended to inform all patients about a treatment the vast majority will not need. Instead, information about biosimilars should be reaching those patients who require such information. In other words, patients who may or will be treated with biosimilars in the near future. This approach differs from informing HCPs about biosimilars, as they all need to have a good understanding of biosimilars.

Educating patients about medicines in general, but in particular biosimilars, should always be a multistakeholder effort [8,20,60,77]. Each stakeholder has its own role to fulfill in order to provide correct, unbiased, understandable, and coherent information. Physicians, nurses, and pharmacists have a coordinating role and are key partners to remove doubts and generate trust in biosimilars, as for any kind of medicine [52,60,85,86]. In addition, other parties such as regulatory authorities, medical societies, and patient associations have a supporting role in informing patients. They are all regarded by patients as reliable sources of information. However, the identified list of stakeholders is not exhaustive, since other stakeholders that were not mentioned in the literature may also play a role. For example, academia might support the development of evidence-based information as a trusted and unbiased source of information. Other national authorities, such as payers and health technology assessment (HTA) bodies, could also disseminate information about biosimilars to patients. Some stakeholders may be of particular importance in the creation of information or educational material (e.g., scientific associations, professional associations, academia), whereas others (e.g., healthcare providers, patient associations, regulatory authorities) in the dissemination of information to patients. Moreover, pharmaceutical companies also play a role in informing the wider public about biosimilar medicines. One must acknowledge that many informational campaigns are supported by pharmaceutical industry, thereby facilitating the development of factual information as well.

### 4.2. The Role of European Patient Organizations

A variety of information and educational material for patients about biosimilar medicines is made public by European patient organizations. Yet, the quality and level of detail vary among different associations, and it is not clear whether the identified information is effectively reaching the patient. This overview of information was based on a web-based screening. However, one should be aware that information made accessible via the internet will not reach every patient who needs such information. After all, not every citizen across Europe has the opportunity to consult the internet. That is why it remains important that healthcare providers fulfill their role to reach patients, and that patient associations themselves do not limit themselves to disseminating information via their websites.

Patient associations often refer to the biosimilar brochure of the European Commission, which was translated in all European languages in recent years. Some patient organizations have developed educational brochures or position statements about the use of biosimilars by themselves. They generally all agree on the fact that biosimilars are equal treatment options ensuring a sustainable healthcare system and underline that the decision to prescribe a biosimilar should be a shared decision involving the patient. Nonetheless, some patient associations should be cautious not to fall prey to negatively framed, incorrect, or outdated information about biosimilars. Several patient associations provide detailed information on biosimilars, but express a rather negative attitude in particular towards transitioning from the reference product to a biosimilar (e.g., IDF Europe, Spanish Platform for Patient Organizations, and IFPA). Others provide or refer to incorrect or outdated information, such as EPDA, IDF Europe, and Flemish Patient Platform. The most pronounced example of this is IDF Europe, where they support their concerns about switching to biosimilar insulins by information that was incorrectly interpreted and taken out of context. Generally, national patient associations adopt the position on biosimilars of their European umbrella organization. However, this does not prevent national associations from formulating their own positions that differ from incorrect European ones. For example, the recommendations of the Dutch Diabetes Association about insulin biosimilars are in line with current scientific evidence and do therefore not correspond to those from IDF Europe [99]. A clear contrast was observed when looking at biosimilar information or educational material of DiCE and National Coalition of Dutch Patients. In particular, DiCE puts emphasis on the fact that if biosimilars are implemented on a wider scale, they could help closing the gap in gaining access to the highest standards of care for the treatment of colorectal cancer. The National Coalition of Dutch Patients repeatedly states that biosimilar medicines have the same efficacy, safety, and quality as their reference products. This is an example of positive framing since most information on biosimilars mentions that no meaningful differences are expected with originator biologicals, which is correct, yet framed more neutrally.

Information should always be evidence-based and therefore in line with the most recent scientific developments. As for all stakeholders, patient associations should distance themselves from positions or opinions about biosimilars that are not scientifically or incorrectly substantiated. Clear collaboration with independent and knowledgeable experts to develop such material is necessary to avoid incorrect information. With this overview, we have taken a critical look at the available information about biosimilars for patients developed by major European patient associations.

### 4.3. Future Perspectives

During past years, the way that most treatment decisions are made has evolved towards shared decision-making [100]. The choice for an originator biological or a biosimilar must therefore be based on a coherent information stream to the patient. Several communication strategies have been identified in this review, guaranteeing correct information is provided adequately to patients. However, not all communication strategies have proven effective in actually increasing patient knowledge and confidence in biosimilars. Moreover, they have not proven to meet the appropriate behavioral objectives among patients. Future research assessing the actual impact of communication strategies based on a behavioral model could help clarify these unmet needs.

Most recommendations identified during this literature review are based on empirical grounds. Communication strategies emerging from theoretical concepts could be explored as well in the future. This would contribute to the overall picture on how to inform patients about biosimilar medicines and increase the robustness of the conclusions.

### 4.4. Strengths and Limitations of the Study

The main conclusions of this study are based on a structured literature review and a web-based mapping of available information by European patient organizations. This study provides an overview of existing scientific literature on how to effectively inform patients about biosimilar medicines. The structured approach allows for reliable conclusions regarding information strategies for patients about biosimilars. This article is the first of its kind to compile the provided information of the major European patient organizations (i.e., EPF and IAPO members), with the purpose to have an overview of available information or educational material.

Although the literature review was conducted in a structured way, no systematic review was conducted and thus the selection of articles was not based on an agreement between two independent researchers. As a consequence, selection bias might have occurred during the title and abstract screening phase. Furthermore, the web-based mapping only allows for the collection of information that is publicly available on the websites of the patient associations of interest. Educational efforts that were not made available on their websites were therefore not included in this review. The researchers chose to include members of EPF and IAPO in the mapping of information, hence some available information on biosimilars by other European patient associations that are not members of these umbrella organizations might have been missed. Although the assessment of the tone in which patient associations report about biosimilars can be seen as subjective, it does provide an interesting picture of the overall attitude of each individual organization and the differences between them.

## 5. Conclusions

It is important to set up a close collaboration between all stakeholders to develop and effectively disseminate correct information about biosimilars to patients, bringing together scientific associations, professional associations (including physicians, nurses, and pharmacists), regulatory authorities, and patient associations. Informing and educating patients on biosimilars should be part of a wider approach to support the adoption of biosimilars in Europe. European member states should consider informing patients on biosimilars in their policy frameworks more actively. It is imperative that European national authorities support biosimilar medicines to safeguard an affordable and sustainable healthcare system within their country.

## Figures and Tables

**Figure 1 pharmaceuticals-14-00117-f001:**
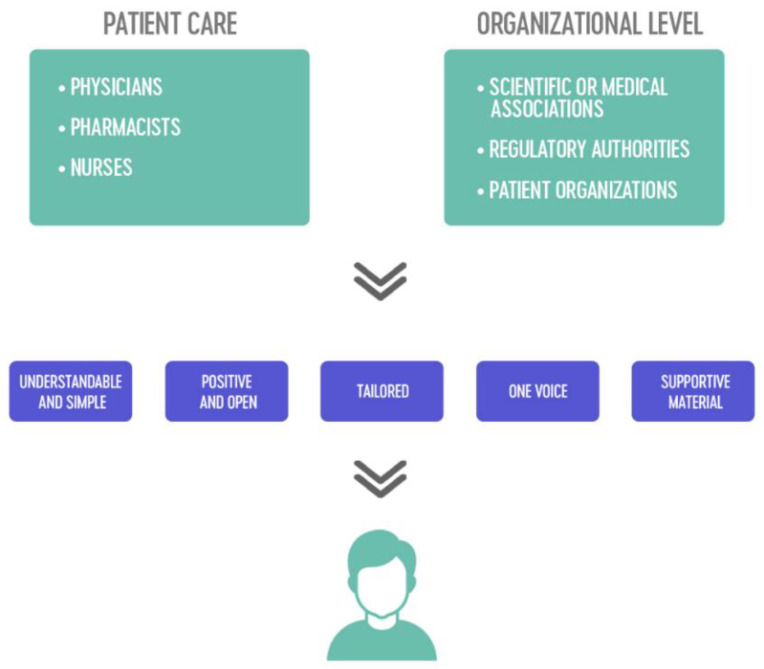
The multistakeholder approach using five main strategies for informing patients about biosimilars.

**Table 1 pharmaceuticals-14-00117-t001:** European Patients’ Forum (EPF) and International Alliance of Patients’ Organizations (IAPO) members providing biosimilar information for patients.

Patient Association	Disease Area	Country/Region of Origin	Available Information	Attitude towards Biosimilars ^1^
Association for the Protection of Patients’ Rights (Asociácia na Ochranu Práv Pacientov, AOPP)	N/A	Slovak Republic	**Short article** about biosimilar medicines (i.e., What are they, how are they produced, the difference with original biologicals) [26]. Lastly, **a link to the EC brochure** (questions and answers about biosimilars for patients) in Slovakian is provided [26].	+
Digestive Cancers Europe (DiCE)	Colorectal cancer	Europe	**Position paper** of DiCE about the use of biosimilar medicines in colorectal cancer (including general information on originator biologicals and biosimilars, biologicals in CRC, access and availability of biologicals, safety and effectiveness of biosimilars). The position paper will be extended to educational materials (video, educational leaflet, checklist to support HCPs) [27]. **General information** on biosimilar and biological medicines, including a **frequently asked questions (FAQ) document [28]**.	+ +
European Federation of Crohn’s and Ulcerative Colitis Associations (EFCCA)	Ulcerative colitis and Crohn’s disease	Europe	**Link to the EC brochure** is provided [29]. **Summary of a workshop** on biosimilars (and biologicals in general) organized by EFCCA [30]. **Short article** on biosimilars in the EFCCA magazine, focusing also on the potential benefits of biosimilar medicines [31].	0
European Multiple Sclerosis Platform (EMSP)	Multiple sclerosis	Europe	**Link to the EC brochure** is provided [32].	+
European Parkinson’s Disease Association (EPDA)	Parkinson’s disease	Europe	**Brief information** on what biological medicines are, with a section on biosimilars. No detailed information is provided [33].	−
International Diabetes Federation European Region (IDF Europe)	Diabetes	International	**Position paper** on biosimilars for the treatment of people with diabetes. This document includes information on the difference with generics (focusing the difference between biosimilars and their reference products), the regulatory framework, impact of biosimilars on healthcare systems, and recommendations for clinical practice [34].	− −
Malta Health Network (MHN)	N/A	Malta	**Link to the EUPATI toolbox** on biosimilar medicines, directed at patients [35].	0
Platform for Patient Organizations (Plataforma de Organizaciones de Pacientes)	N/A	Spain	**Specific web page about biological medicines in general and biosimilars**, including information on their definitions, interchangeability, substitution, and position statements [36].	−
European Federation of Neurological Associations (EFNA)	Neurological disorders	Europe	**Link to the EC brochure** is provided [37].	+
European Institute of Women’s Health (EIWH)	N/A	Europe	**Link to the EC brochure** is provided [38].	+
National Coalition of Dutch Patients (Patiëntenfederatie Nederland)	N/A	Netherlands	**Brief information** on key concepts of biosimilars [39]. **Link to a brochure (question and answer)** about biosimilars developed by the Dutch competent authority, including general information, their position on interchangeability and switching, and infographics about biosimilar medicines [40].	+ +
Flemish Patient Platform (Vlaams Patiëntenplatform) (FPP)	N/A	Belgium	**Very brief information** on biological and biosimilar medicines (‘copy of original biological, equal to generics’) [41]. **Link to specific information** from the Belgian regulatory authority is provided. This information includes: definition, general information, pharmacovigilance, available biosimilars (not up to date), and links to several other brochures (EC, EMA, etc.) [42].	−
European Patients’ Forum (EPF)	N/A	Europe	**A link to the EC brochure** is provided. Several EPF members collaborated with EC and EMA on the EC brochure about biosimilar medicines for patients [43]. A **summary of the yearly biosimilar stakeholder** event by the EC [44].	0
International Federation of Psoriasis Associations (IFPA)	Psoriasis	International	**Position statement** on the use of biosimilar medicines for the treatment of psoriasis, including the definition, general information, switching, regulatory requirements [45].	−
Psoriasis Action (Acción Psoriasis)	Psoriasis	Spain	**Link to a video** where biosimilars are explained by an expert [46]. **Short article about biosimilar medicines**, explaining general information about them [47,48].	0
International Alliance of Patients’ Organizations (IAPO)	N/A	International	**Biosimilars toolkit,** developed in collaboration with IFPMA, is publicly available on the IAPO website. The toolkit contains information on several aspects of biologicals in general, and biosimilar medicines specifically: general information, regulatory requirements, pharmacovigilance, how to talk to patients about biosimilars, biologicals in low- and middle-income countries, key recommendations (as mentioned by WHO), and FAQs about biosimilars [49].	0

^1^ The evaluation of the overall attitude towards biosimilars for each patient organization is done on a five-point Likert scale. The scale is as follows: “− −” (negative), “−” (somewhat negative), 0 (neutral), “+” (somewhat positive), and “+ +” (positive).

## Data Availability

The data presented in this study are available on request from the corresponding author.

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
