# Peer review of "Informing Patients about Biosimilar Medicines: The Role of European Patient Associations"

_pharmaceuticals, 2021, doi:10.3390/ph14020117_

Round 1

Reviewer 1 Report

This is a well-written paper about an interesting and relevant topic. It provides relevant information and suggestions for policy and practice. Yet, I also have some major comments, mainly about methods and thus about the soundness of the recommendations.

There is a lot of literature about how to communicate about medication with patients. Many of the suggestions provided are general suggestions for medicines. I would have preferred a more theory-based approach. I can imagine you are not going to do that at this stage, but I'd mention it either as limitation of as a strong recommendation, to embedd suggestions not only on empirical grounds but also to use theoretical concepts to map information needs.

My main concern about this paper is the methods that are used. They are very poorly described. My main criterion is that I would be able to repeat the study after readlng the methods section. I would not be able to do that for this study. In the appendix (which I could not download) there only seems to be an GANTT chart. I'd  add also the exact search strategy. Also, no information is given about how papers were selected: what were in- and exclusion criteria? Also, for a sound review, two people need to do the screening of abstracts and full texts. Was this the case here? If yes, how was the level af agreement? How were conflicts betweend the two reviewers resolved? And if only one person did the selection, why was that and how could that have impacted the results? What data were extracted and how were the papers analysed? Did you have open coding? Or did you have ideas about the topics you wanted to address. It is all very unclear. Even for a more narrative review like this more information is needed. Also I would look at changes over time in ideas about information on biosimilasrs. As they were new in 2006, I can imagine, communcation would be different as less was known about wide application of these biosimilars, so patients might have been more uncertain than they are now, where they know many other patients have had a biosimilar.

Why were abstracts included? Sometimes they hold preliminary results.

Also, the search on the websites could be described more thoroughly. How did you search and also, how was the information evaluated. You put quite some strong statements about the quality of the information. And I do not mind that you do that, but against what standard was it evaluated? There is no mention about that in the methods section.

The results read very well and are interesting. Still, I can not value them fully given the methodological limitations. The results of the websites of patient organization could be shortened. I'd put more information in the table, describing more there and provide a more overarching and shorter analysis in the main text.

In the discussion, which again is very well written and interesting to read I miss a reflection on the strenghts and limitations of this review.

So, all in all, this a review that is worth publishing but needs to be more critical towards its own methods and reflections.

Author Response

Dear reviewer, 

Many thanks for your comments and suggestions regarding this manuscript entitled “Informing patients about biosimilar medicines: The role of European patient associations”. 

All detailed revisions per line are included in the updated cover letter (Cfr. attachment). Below you can find our responses to your remarks or suggestions, point by point:

1. There is a lot of literature about how to communicate about medication with patients. Many of the suggestions provided are general suggestions for medicines. I would have preferred a more theory-based approach. I can imagine you are not going to do that at this stage, but I'd mention it either as limitation of as a strong recommendation, to embedd suggestions not only on empirical grounds but also to use theoretical concepts to map information needs.

Response: We agree that most of our suggestions to communicate effectively about biosimilars are indeed applicable to other medicines as well. However, we believe that a structured and comprehensive review of existing literature on how to inform patients (about biosimilars) was the most valid approach to identify the main points to consider when informing patients on biosimilars. Some of these articles did include general theoretical concepts as well, in addition to empirical evidence. Yet, we have added the suggestion for further work to include theoretical frameworks to formulate communication strategies. We believe that this would contribute to the robustness of the formulated conclusions. We added a section under the discussion on future perspectives, this point was described there. 

2. My main concern about this paper is the methods that are used. They are very poorly described. My main criterion is that I would be able to repeat the study after reading the methods section. I would not be able to do that for this study. In the appendix (which I could not download) there only seems to be a GANTT chart. I'd add also the exact search strategy. Also, no information is given about how papers were selected: what were in- and exclusion criteria? 

Response: Thank you for this remark and suggestion. The authors agree with your remarks about the description of the methods, that they were not described in detail in this version. Therefore, we have elaborated on the methods used in the methods section. The appendix now includes the complete search strategy with the search query and inclusion criteria, in addition to the PRISMA flow chart.

3. Also, for a sound review, two people need to do the screening of abstracts and full texts. Was this the case here? If yes, how was the level of agreement? How were conflicts between the two reviewers resolved? And if only one person did the selection, why was that and how could that have impacted the results?

Response: Thank you for this remark, we agree that this was not well clarified in the manuscript. We conducted a structured and comprehensive literature review. We did not conduct a systematic literature review, which would indeed include two people reviewing and screening the abstracts/full text of the articles. We have chosen not to conduct a systematic literature review, because the main purpose of this article was to provide a mapping of available information and include overall communication strategies that were identified from the scientific literature. Therefore, a complete systematic review with an independent screening of two researchers was not deemed necessary. The authors agree that this is a limitation and could have lead to selection bias. Hence, this was included as a limitation in the manuscript, that was added under the discussion. 

4. What data were extracted and how were the papers analyzed? Did you have open coding? Or did you have ideas about the topics you wanted to address? It is all very unclear. Even for a more narrative review like this more information is needed. 

Response: Thank you for this question, it was indeed not very well clarified in the manuscript. We clarified the overall process of data analysis in the methods section of the manuscript. The articles included in the final analysis were analyzed qualitatively according to the thematic framework method. A combination of inductive and deductive coding was used since some aspects were already identified as relevant for this research question. During the initial coding step, general themes were identified prior to the literature review. Similar codes were grouped together to form the coding tree. Second, the identified literature was coded deductively. Meanwhile, additional codes were created inductively and added to the coding tree.

5. Why were abstracts included? Sometimes they hold preliminary results.

Response: Abstracts were included for the completeness of this comprehensive review. Although some of them indeed hold preliminary results, the researchers aimed to include all existing evidence on communication strategies about biosimilar medicines. 

6. Also, the search on the websites could be described more thoroughly. How did you search and also, how was the information evaluated. You put quite some strong statements about the quality of the information. And I do not mind that you do that, but against what standard was it evaluated? There is no mention about that in the methods section.

Response: Thank you for this suggestion. We have included additional information on how the web-based screening has been performed exactly. This was added to the methods section as well. We acknowledge the limitation of our evaluation of the tone or attitude on how patient associations report about biosimilars. This was also included under the added limitations section in the discussion. However, the authors believe that such an assessment provides an interesting view on how different associations report about biosimilars, and provide an indication of possible differences among patient associations. 

7. The results read very well and are interesting. Still, I can not value them fully given the methodological limitations. The results of the websites of patient organization could be shortened. I'd put more information in the table, describing more there and provide a more overarching and shorter analysis in the main text.

Response: For the overall readability of the manuscript, we indeed chose to include only brief information in the table and describe the results more elaborately throughout the text. We think that the table would become less organized if more information would be added. 

8. In the discussion, which again is very well written and interesting to read I miss a reflection on the strengths and limitations of this review.

Response: Thank you for this remark. The authors fully agree on the fact that the limitations of the chosen methods should have been clarified. We added a section on strengths and limitations to the discussion. By this, the overall results and recommendations can be valued better taking into account the methodological limitations. All specific changes to the manuscript can be found in the updated cover letter, as well as in the revised manuscript as track changes, for your convenience. 

Thank you in advance for considering this revised version of the manuscript.  

Kind regards,

Yannick Vandenplas (on behalf of all co-authors)

Reviewer 2 Report

Ms: Pharmaceuticals-1080408 “Informing patients about biosimilar medicines: The role of European patient associations”

The authors map and analyse the biosimilars’ information available for patients on web-pages/papers. They identify a highly heterogeneous message throughout them, and propose some measures to optimize the delivery of facts and reassure patients on their use. They conclude on the need for a single voice from all stakeholders to deliver correct and efficient information in addition to providing other proposals.

General Comment

This is a very timely and interesting manuscript with an overly amount of work/information. I believe that no such type of extensive review has ever been published. It is a remarkable piece of work that will need to me further scrutinized to give rise to further works that take may take various directions.

General Comments

  • Although this reviewer has no major comments/remarks, he wishes to highlight two issues that are worth being touched upon (although I may understand the reasons for not doing so):
  • Industry (innovator and biosimilar) may play a role in all this They are one of the market stakeholders and yet minor attention has been given to them. It may be worth briefly addressing their potential beneficial role in conveying positive messages (at least biosimilar companies). As matter of fact most of the education material is sponsored directly or indirectly by industry.
  • It would be extremely helpful to provide specific examples of misleading or wrong information by specific papers/web pages. Rather than just deliver a general discussion, such examples will help readers specifically understand the authors identified pitfalls
  • On the other hand, yet probably not easy, it may be worth give an example of a positive adequate message for patients. Given all the analysed info/facts, a proposal of a short message simulating physician-to-patient communication on biosimilar when facing a switching decision for instance. I am not sure that this is easy appropriate but I leave the thought there for the authors consideration.

Specific Comment

  • The authors claim that “A complete overview of the literature search process is included in Supplementary Figure S1”. I haven’t actually found that info on Fig 1

  • IMPORTANT

As part of the proper/accurate communication, please avoid expressions such as “…biological and biosimilar medicines…”. This actually creates the idea that a biosimilar is not a biological medicine, yet as the authors very well know, it is. So do please either differentiate between “original biologics and biosimilars” or “off-patent biologics and biosimilars” (I would go for the former). On a few occasions throughout the text biologis and biosimilars medicines are wrongly confronted.

Author Response

Dear reviewer, 

Many thanks for your comments and suggestions regarding this manuscript entitled “Informing patients about biosimilar medicines: The role of European patient associations”. 

All detailed revisions per line are included in the updated cover letter (Cfr. attachment). Below you can find our responses to your remarks or suggestions, point by point:

1. This is a very timely and interesting manuscript with an overly amount of work/information. I believe that no such type of extensive review has ever been published. It is a remarkable piece of work that will need to me further scrutinized to give rise to further works that take may take various directions.

Response: Many thanks for your reaction to our manuscript. 

2. Industry (innovator and biosimilar) may play a role in all this They are one of the market stakeholders and yet minor attention has been given to them. It may be worth briefly addressing their potentially beneficial role in conveying positive messages (at least biosimilar companies). As matter of fact, most of the education material is sponsored directly or indirectly by industry.

Response: Thank you for this remark. The role of the pharmaceutical industry when informing patients about biosimilars was added to the manuscript in the discussion section. We agree that they have the responsibility to communicate correctly about biosimilar medicines as well. However, their role in reaching out to patients directly is rather limited since this is not allowed in many European countries. The authors agree that we should acknowledge that many educational campaigns, especially by patient organizations, are funded by the industry. 

3. It would be extremely helpful to provide specific examples of misleading or wrong information by specific papers/web pages. Rather than just deliver a general discussion, such examples will help readers specifically understand the authors identified pitfalls. On the other hand, yet probably not easy, it may be worth giving an example of a positive adequate message for patients. Given all the analyzed info/facts, a proposal of a short message simulating physician-to-patient communication on biosimilar when facing a switching decision for instance. I am not sure that this is easily appropriate but I leave the thought there for the authors' consideration.

Response: It is indeed useful to include clear examples of positive and negative messages given by the patient associations. The authors believe that the more negative messages were addressed in the results section and mentioned again in the discussion. However, more details on the positive messages of DiCE and the National Coalition of Dutch Patients are added in the discussion. In particular, the difference between stating that no differences in clinical outcomes are expected after transitioning from a reference biological to a biosimilar, and, the same clinical outcomes are expected. This is an example of positive framing, which is known to benefit the confidence of patients in biosimilars.

4. The authors claim that “A complete overview of the literature search process is included in Supplementary Figure S1”. I haven’t actually found that info on Fig 1. 

Response: Thank you for this comment. In the first version, we only included the PRISMA flow chart of the structured literature review. In the revised version, we added the complete search strategy including the search query and inclusion criteria. 

5. As part of the proper/accurate communication, please avoid expressions such as “…biological and biosimilar medicines…”. This actually creates the idea that a biosimilar is not a biological medicine, yet as the authors very well know, it is. So do please either differentiate between “original biologics and biosimilars” or “off-patent biologics and biosimilars” (I would go for the former). On a few occasions throughout the text, biologics and biosimilars medicines are wrongly confronted.

Response: Many thanks for this remark. We can only agree that it is paramount to correctly refer to these terms and avoid confusion as much as possible. We have made adjustments at several places throughout the manuscript where confusion might occur, leading to the misinterpretation that biosimilars are no biological medicines. The exact changes (and where) can be found in the revised manuscript and cover letter, for your convenience. 

Thank you in advance for considering this revised version of the manuscript.  

Kind regards,

Yannick Vandenplas (on behalf of all co-authors)

Reviewer 3 Report

The review aims to provide an overview of existing scientific literature on how to inform patients about biosimilars and available information about biosimilars for patients, developed or disseminated by European patient associations. 

As the authors mention, a multitude of studies have been conducted on
knowledge about or attitudes towards biosimilar medicines among patients, indicating that relevant information or relevant ways to convert the information have not been sufficiently utilised yet, supporting the relevance of the research questions.

The research is using a comprehensive literature review and web based screening of informations available for patients. While the literature review is just introduced as comprehensive, it seems to include core components of a systematic literature review. Clarification would be helpful, as only a systematic literature review would allow for the stronger conclusions. 

Both research steps are identifying currently provided information and perspective on the information important to patients. The five main points of attention identified seem to have face validity. However, they are based on the current activities and believes in informing patients which do not seem to appropriately meet the behavioural objectives. This would be a relevant aspect for the discussion as well demonstrate the main weakness of the descriptive methodological approach. Only a behavioural model could provide insights on how to revise the current approach of informing patients to elevate the effectiveness and efficiency in achievement of the behavioural objectives.

Author Response

Dear reviewer, 

Many thanks for your comments and suggestions regarding this manuscript entitled “Informing patients about biosimilar medicines: The role of European patient associations”. 

All detailed revisions per line are included in the updated cover letter (Cfr. attachment). Below you can find our responses to your remarks or suggestions, point by point:

1. The research is using a comprehensive literature review and web-based screening of information available for patients. While the literature review is just introduced as comprehensive, it seems to include core components of a systematic literature review. Clarification would be helpful, as only a systematic literature review would allow for stronger conclusions. 

Response: Many thanks for this remark. We agree that we did not clear enough what the structured literature review entailed. This has been added to the methods section, including more details on the methodological aspects.

Since it was not very clear whether this review concerned a systematic literature review or not, additional explanations were added to the methods section about the literature review (2.1). The comprehensive literature review indeed addresses the research question based on a structured search strategy, which has also been added now to the supplementary material (Cfr. Table S1). However, it should not be regarded as a systematic literature review since the screening step has not been performed by two independent researchers. The purpose of this paper was to provide a comprehensive literature review, based on a structured search strategy.

Furthermore, we acknowledge that a comprehensive structured literature review has some limitations in comparison to a systematic literature review (i.e. selection bias). We have added a section on the study limitations under the discussion, where we address these methodological limitations. 

2. Both research steps are identifying currently provided information and perspective on the information important to patients. The five main points of attention identified seem to have face validity. However, they are based on the current activities and believes in informing patients which do not seem to appropriately meet the behavioral objectives. This would be a relevant aspect for the discussion as well as demonstrate the main weakness of the descriptive methodological approach. Only a behavioral model could provide insights on how to revise the current approach of informing patients to elevate the effectiveness and efficiency in the achievement of the behavioral objectives.

Response: Thank you for this remark. We agree that many of the identified articles that report on communication strategies did not prove to meet the behavioral objectives. The importance of further research on behavioral objectives, based on a behavioral model, was added to the discussion section as well. We think that this is a valuable suggestion for further research in this particular field. Therefore, we added a specific section on future perspectives under the discussion where we address this topic as well. All specific changes to the manuscript can be found in the updated cover letter, as well as in the revised manuscript as track changes, for your convenience. 

Thank you in advance for considering this revised version of the manuscript.  

Kind regards,

Yannick Vandenplas (on behalf of all co-authors)

Round 2

Reviewer 1 Report

I approve of the changes made by the authors. I recognize some of my remarks were not easy to adpat in this stage but these are sufficiently addressed in the discussion section. I have no further comments.